# Efficient gene knockout and genetic interaction screening using the in4mer CRISPR/Cas12a multiplex knockout platform

Nazanin Esmaeili Anvar [1,2,6], Chenchu Lin[1,6], Xingdi Ma[1,2,6], Lori L. Wilson[1], Ryan Steger[3], Annabel K. Sangree[3], Medina Colic[1], Sidney H. Wang[4], John G. Doench [3] & Traver Hart [1,5] ✉

Genetic interactions mediate the emergence of phenotype from genotype, but technologies for combinatorial genetic perturbation in mammalian cells are challenging to scale. Here, we identify background-independent paralog synthetic lethals from previous CRISPR genetic interaction screens, and find that the Cas12a platform provides superior sensitivity and assay replicability. We develop the in4mer Cas12a platform that uses arrays of four independent guide RNAs targeting the same or different genes. We construct a genome-scale library, Inzolia, that is ~30% smaller than a typical CRISPR/Cas9 library while also targeting ~4000 paralog pairs. Screens in cancer cells demonstrate discrimination of core and context-dependent essential genes similar to that of CRISPR/Cas9 libraries, as well as detection of synthetic lethal and masking/buffering genetic interactions between paralogs of various family sizes. Importantly, the in4mer platform offers a fivefold reduction in library size compared to other genetic interaction methods, substantially reducing the cost and effort required for these assays.

Pooled library CRISPR screens have revolutionized mammalian functional genomics. DepMap teams have screened over a thousand cancer cell lines with CRISPR knockout libraries to identify background-specific genetic vulnerabilities[1–3], while dozens of genetic modifier screens with small molecules have explored biomarkers and mechanisms of drug sensitivity and resistance[4–10]. However, initial efforts to assay genetic interactions (GIs) – that is, the manipulation of multiple genes in the same cell to identify nonlinear combinatorial phenotypes – have proven complex and costly[11–15]. One class of GIs that has received special attention is the synthetic lethal relationship between paralogs, gene pairs or families that arise through duplication of a single ancestral gene. Functional buffering by paralogs, resulting in phenotypic masking in single gene perturbation experiments, has

been shown extensively in model organisms[16,17]. Paralogs are therefore attractive targets for genetic interaction studies in human cells, and they are more easily nominated by computational analyses[18,19] compared to genes that work in parallel pathways, such as *BRCA1* and *PARP1*[20]. Further, because the mechanism of action of drugs often relies on inhibition of paralog gene products to mediate cell toxicity, monogenic knockout in CRISPR screens have resulted in false negatives, such as the failure to identify MEK and ERK proteins as critical for cancer cell growth.

Recently, paralog synthetic lethals have been assessed with multiple CRISPR-based methods, with some relying on a single Cas endonuclease with multiple guides and others using orthogonal Cas proteins in the same cell[19,21–24]. However, with the application of different

[1]Department of Bioinformatics and Computational Biology, The University of Texas MD Anderson Cancer Center, Houston, TX, USA. [2]Graduate School of Biomedical Sciences, The University of Texas MD Anderson Cancer Center UTHealth, Houston, TX, USA. [3]Genetic Perturbation Platform, Broad Institute of MIT and Harvard, Cambridge, MA, USA. [4]Center for Human Genetics, The Brown foundation Institute of Molecular Medicine, The University of Texas Health Science Center at Houston, Houston, TX, USA. [5]Department of Cancer Biology, The University of Texas MD Anderson Cancer Center, Houston, TX, USA. [6]These authors contributed equally: Nazanin Esmaeili Anvar, Chenchu Lin, Xingdi Ma. ✉e-mail: traver@hart-lab.org

experimental and informatic pipelines, comparison across studies has been challenging. Notably, no widely accepted gold standard set of paralog synthetic lethals exists against which researchers can calculate traditional metrics of accuracy such as sensitivity and specificity.

Here, we describe a meta-analysis of paralog genetic interaction screens in human cells, identifying a set of background-independent paralog synthetic lethals, and demonstrate that the enhanced version of Cas12a[25] from *Acidaminococcus sp.* (enAsCas12a[26], hereafter referred to as Cas12a) provides the best combination of sensitivity and simplicity for genetic interaction studies. Building on our prior work[19,27,28], we further extend the capabilities of the Cas12a system. We show that it can reliably utilize arrays encoding four optimized gRNAs when delivered by lentivirus in a pooled screening format. We describe the in4mer platform that uses oligonucleotide synthesis to construct libraries of four-guide arrays that target specified sets of one to four genes independently. From the in4mer platform, we develop a genome-scale human library named Inzolia which, with ~49,000 clones, is ~30% smaller than standard whole-genome libraries and has the unique added capability of targeting more than 4000 paralog families of size two, three, and four.

## Results

### Comparing dual-gene knockout studies and identifying synthetic lethal interactions

With the discovery that paralogs are both systematically underrepresented as hits in pooled library screens and likely offer the highest density of genetic interactions, several independent studies have each targeted hundreds of paralog pairs in multiple cell lines[19,21–24]. However, evaluating the quality and consistency of these studies has proven difficult, since each uses a different technology and custom analytics pipeline for hit calling, and overlap between the targeted paralog pairs in each study is surprisingly slim (Fig. 1A, B).

We developed a unified genetic interaction calling pipeline, based on measuring a pairwise gene knockout's deviation from expected phenotype (delta log fold change, dLFC) and the standardized effect size of this deviation (Cohen's d) (Fig. 1C, D and Supplementary Fig. 1). After performing background-specific normalization (see "Methods" section), we classified paralogs as synthetic lethal if they met both dLFC and Cohen's d thresholds ("Methods" section). A total of 388 gene pairs were scored as hits across all five multiplex perturbation platforms (Fig. 1E).

Using this pipeline, we found the large majority of paralog synthetic lethals to be platform-specific. To aid in comparing hits within and across experiments, we developed a platform quality score that broadly measures the replicability of these synthetic lethal screening technologies across different cell lines. We reasoned that, like individual essential genes, a large fraction of paralogs should show consistent synthetic lethality across most or all cell lines, which should be reflected in similarity of synthetic lethality profiles across cell lines. We therefore calculated the Jaccard coefficient of each pair of cell lines screened by a particular platform, then took the median of each platform's Jaccard coefficients as the platform quality score ("Methods" section and Fig. 1F).

We then calculated a paralog confidence score for each gene pair by taking the sum of each hit, weighted by the platform quality score, and subtracting the sum of each experiment in which the pair was assayed but not deemed a hit (a "miss"), also weighted by quality score (Fig. 1G). Using this approach, paralog pairs that are hits in multiple high-quality screens outweigh pairs that are hits in screens with lower replicability or pairs that are background-specific hits in high-scoring screens. We further filtered for hits that are detected by more than one platform, minimizing the bias toward paralog pairs that are only assayed in one set of screens or with one perturbation technology. We identified a total of 26 gene pairs that meet these criteria, and we classified the top 13 hits (with paralog score > 0.25) as candidate

paralog synthetic lethal gold standards (Fig. 1H–J and Supplementary Data 1). Measuring the recall of each of the 21 cell line screens against these gold standards confirmed that the Cas12a platform used in Dede et al.[19], with two Cas12a guide RNAs expressed from the same promoter, yielded the highest within-platform replicability (Supplementary Fig. 2). Other platforms often showed high sensitivity in one screen, but highly variable sensitivity across multiple screens (Supplementary Fig. 2).

We note that our approach does not consider the experimental designs used in each of the five paralog synthetic lethality studies. Differences in the set of chosen paralogs, library coverage, experimental timepoint, and sequencing depth, for example, might be sources of variance among each of the studies. We assumed that research teams applied best practices to their studies, but it is also possible that experimental design, rather than intrinsic platform robustness, dominates the within-study variation that we observed.

### Optimizing the Cas12a system for multiplex perturbations

Based on the consistency of the Cas12a results in the paralog screens and its potential applications to higher-order multiplexing, we explored whether crRNA arrays longer than two guides could be utilized at scale. The Cas12a system has previously been shown to mediate multiplexing beyond two targets[23,29,30] with varying levels of efficiency. Guide RNA design is a critical factor in all CRISPR applications[31], and empirical data on Cas12a guide efficacy is relatively sparse compared to >1000 whole-genome screens in cancer cell lines performed with Cas9 libraries. We tested more than 1000 crRNA from the CRISPick[27,32] design tool in a pooled library targeting coding exons of known essential genes and found very strong concordance between the CRISPick on-target score and the fold change induced by the gRNA (Supplementary Fig. 3). We therefore considered CRISPick designs for all subsequent work.

We have previously shown that arrays encoding two crRNAs show minimal position effects[19,27,28], but information about longer arrays is sparse[23,29,30]. We constructed arrays of up to 7 gRNAs to evaluate the maximum length that would yield gene knockout efficiency sufficient for pooled library negative selection screening. A set of seven essential and non-essential genes were selected and each assigned to one position (1–7) on the array. A single-guide RNA was selected for each gene, and arrays were constructed such that all combinations of essential and non-essential gRNA were represented, for a total library diversity of 128 array sequences (Fig. 2A). The process was repeated two more times, using different gRNA targeting the same genes, creating three pools each of 128 unique sequences, each targeting the same seven essential and non-essential genes in all combinations (Fig. 2B).

We cloned the 7mer pools into the pRDA_550 vector, a one-component lentiviral vector expressing the Cas12a CRISPR endonuclease and the *pac* puromycin resistance gene from an EF-1α promoter and the array of Cas12a guides from a human U6 promoter (see Methods). We used the library to screen K-562 cells, a *BCR-ABL* chronic myeloid leukemia cell line commonly used for functional genomics, and collecting samples at 7, 14, and 21 days (Fig. 2C). After normalization (see Methods), arrays with no essential gRNA showed no sign of negative selection compared to arrays with any number of essential gRNA. Arrays with multiple essential guides showed increasing loss of fitness, reaching maximal phenotype at 4 to 5 essential guides per array (Fig. 2D; Supplementary Fig. 4; Supplementary Data 2; and Supplementary Data 3).

To evaluate position-level effects, we considered arrays encoding a single essential gRNA at any of the seven positions. Across the three replicates, we consistently observed greater fold change at the first four positions compared to the last three positions on the array (Fig. 2E). We further tested whether this efficiency drop at the end of the array was a position-dependent effect or the result of unfortunate

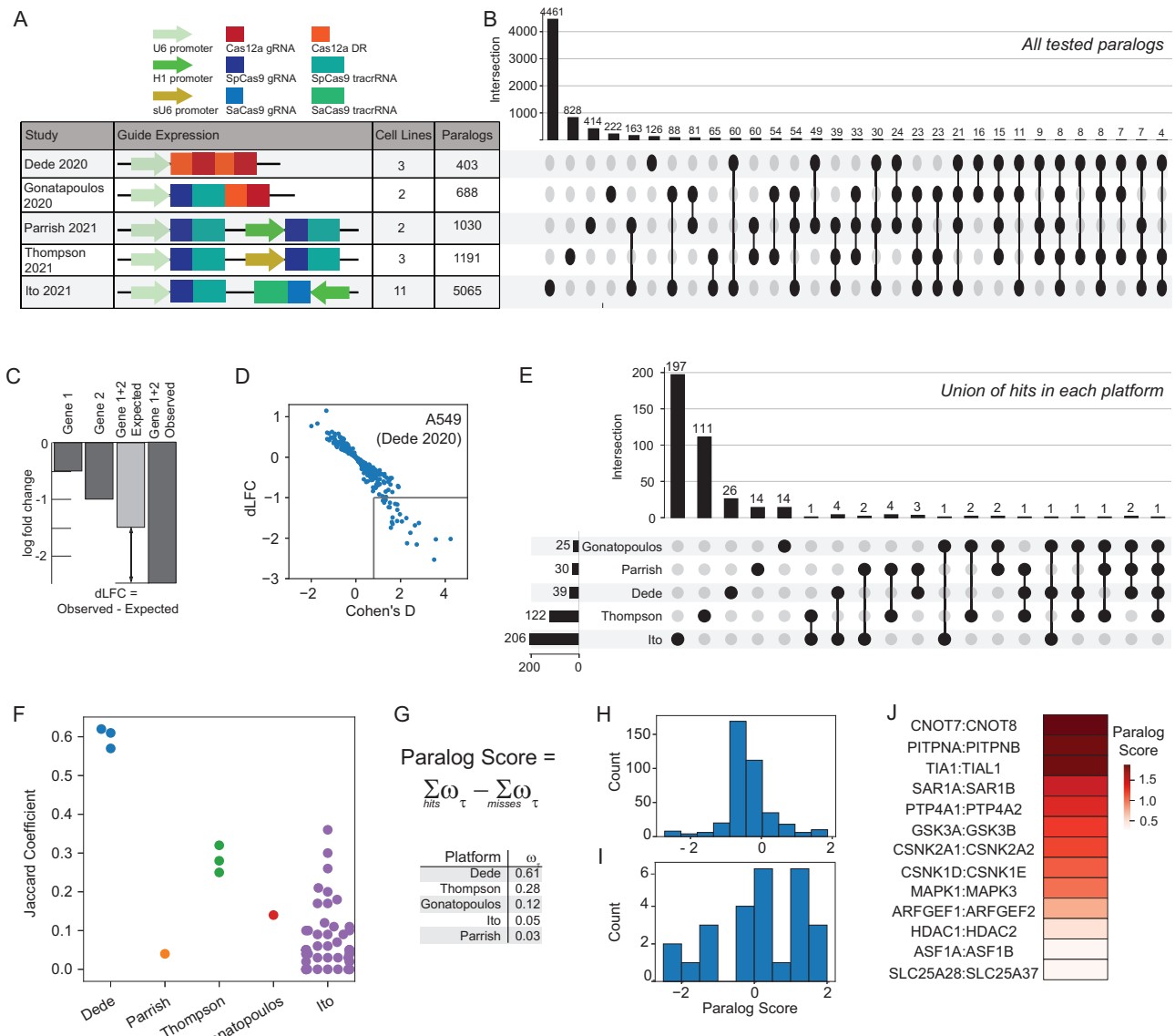

**Fig. 1 | Comparative analysis of paralog synthetic lethality screens. A** Different multiplex CRISPR perturbation methods applied to assay paralog synthetic lethality. **B** Tested paralog pairs in each study. Upset plot shows the intersection of pairs across different studies. **C** Quantifying synthetic lethality between paralog pairs. Single mutant fitness (SMF) is the mean log fold change of gRNAs that target an individual gene. Expected double mutant fitness (DMF) is calculated as the sum of SMF of gene 1 and gene 2. Delta Log Fold Change (dLFC) is the difference between observed and expected fold change and is used as a measure of genetic interaction. **D** dLFC vs. Cohen's D in one data set, A549 screen in Dede.

**E** Comparison of union of hits across all cell lines in each study. **F** Jaccard coefficient comparing hits across all pairs of cell lines within each study. **G** The "paralog score" is the weighted sum of hits minus the weighted sum of misses; i.e. where the gene pair is assayed but not a hit. Weights are the median of the platform-level Jaccard coefficients from **F**. **H** Histogram of paralog scores of 388 hits across all 5 studies. **I** Histogram of paralog scores across 26 hits in >1 study. **J** Thirteen candidate "paralog gold standards" with paralog score >0.25 and hit in more than one study. Source data are provided as a Source Data file.

guide or gene selection. We constructed a second array with the same gRNA targeting the same genes in reverse order (one essential gRNA per array) and re-screened the same cells. When comparing the fold change of the forward array with the reverse array, observed fold changes on the diagonal indicate gene- and guide-level effects independent of position, while deviations from the diagonal indicate position-specific effects. Our data confirm that the first four to five gRNAs show no position-specific effects, but positions six and seven show marked deviation from the diagonal (Fig. 2F). Based on these observations, we conservatively conclude that the Cas12a system using the pRDA_550 vector can effectively express and utilize arrays of four gRNAs.

We also evaluated whether the 7mer array could be used to identify combinatorial phenotypes. We trained a linear regression model using a binary encoding of guide arrays as a predictor (where non-essential = 0 and essential = 1) and log fold change as a response variable (see Methods). The regression model provides excellent prediction of fold change for arrays encoding two essentials ($R^2 = 0.78$–$0.91$ for the three pools) from the sum of calculated single-guide position-level regression coefficients (Fig. 2G). These observations are consistent with the multiplicative model of genetic interactions, which predicts that the result of independent loss of fitness perturbations is the sum (in log space) of the fold changes of the individual fitness perturbations. It further supports the utility of the Cas12a platform for multiplex perturbation and detection of genetic interactions, which are simply deviations from the expected phenotype according to this model, because the null model accurately fits the data for independent combinatorial perturbations.

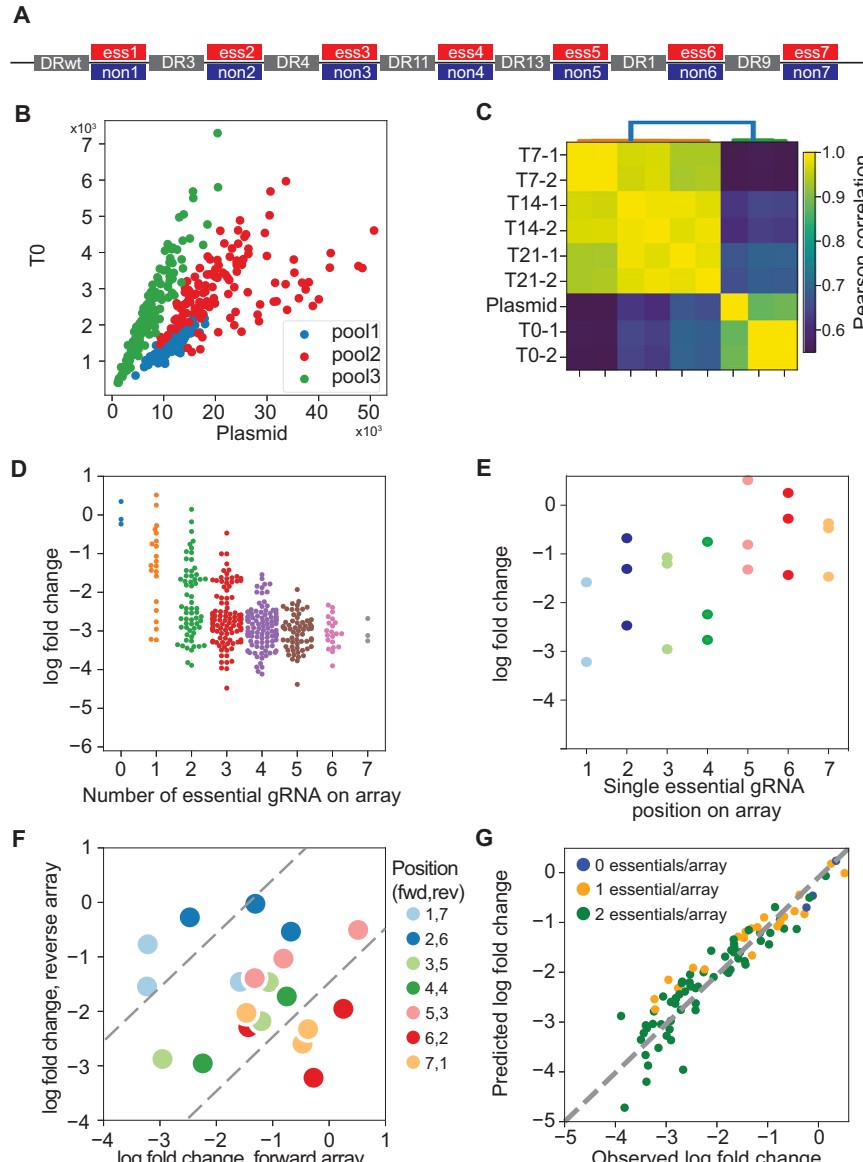

**Fig. 2 | Multiplexing beyond 2 guides with Cas12a. A** 7mer arrays were constructed with all combinations of either an essential or non-essential guide at each position ($2^7 = 128$ species), using the same DR sequences at each position, in three independent sets with unique gRNA sequences targeting the same genes at each position ($n = 384$ total). **B** Guide sets were evenly represented in the combined pool before and after packaging and transduction (**C**) 7mer guide array representation is consistent across replicates and variation is consistent with high quality screens. **D** Fold change of guide arrays vs. number of essential guides on the array ($n = 384$

arrays). **E** Fold change vs. position of essential guide on array, for all arrays encoding one essential guide (six nonessentials). **F** Fold change of guide arrays encoding one essential per array, forward vs. reverse orientation. Essential guides expressed at positions 6 and 7 deviate from the diagonal, indicating position-specific loss of editing. **G** A linear regression model that learns single gene knockout effects can be used to predict combinatorial target phenotype, an accurate null model for genetic interactions.

## The in4mer platform for single and combinatorial perturbation

With confidence that the Cas12a platform supports independent utilization of four guides expressed from a single array, we designed a prototype genome-scale library that targets both protein-coding genes and paralog families in the same pool. Each array encodes four distinct gRNA, each with a different DR sequence selected from the top performers in DeWeirdt et al.[27] (Fig. 3A). The prototype library targets each of 19,687 protein-coding genes with one four-guide array encoding 20mer crRNA sequences from the top four guides nominated by the CRISPick algorithm, and a second four-guide array using the same guides in a different order. This initial library also targets 2082 paralog pairs with a single array encoding two gRNA per gene and a second array encoding the same gRNA in a different order (see Methods and Supplementary Fig. 5 for paralog selection strategy, see

Supplementary Data 4 for gRNA arrays sequence). Additionally, 167 paralog triples and 48 paralog quads are targeted by two arrays, with each array encoding a single-guide targeting each gene. Arrays targeting triples are padded with a fourth guide targeting a randomly selected non-essential gene. For triples and quads, the two arrays per set encode different gRNA sequences (Fig. 3A). Total prototype library size is 43,972 4mer CRISPR arrays, including 4 arrays with 4 guides each targeting EGFP. Since the leading direct repeat sequence is already on the pRDA_550 backbone, the library can be synthesized as a 212mer oligo pool, including 5' and 3' amplification and cloning sequences (see "Methods" section).

We conducted screens in K-562, a *BCR-ABL* chronic myeloid leukemia cell line and in A549, a *KRAS*-mutant lung cancer cell line with wildtype *TP53*, using standard CRISPR screening protocols (500x

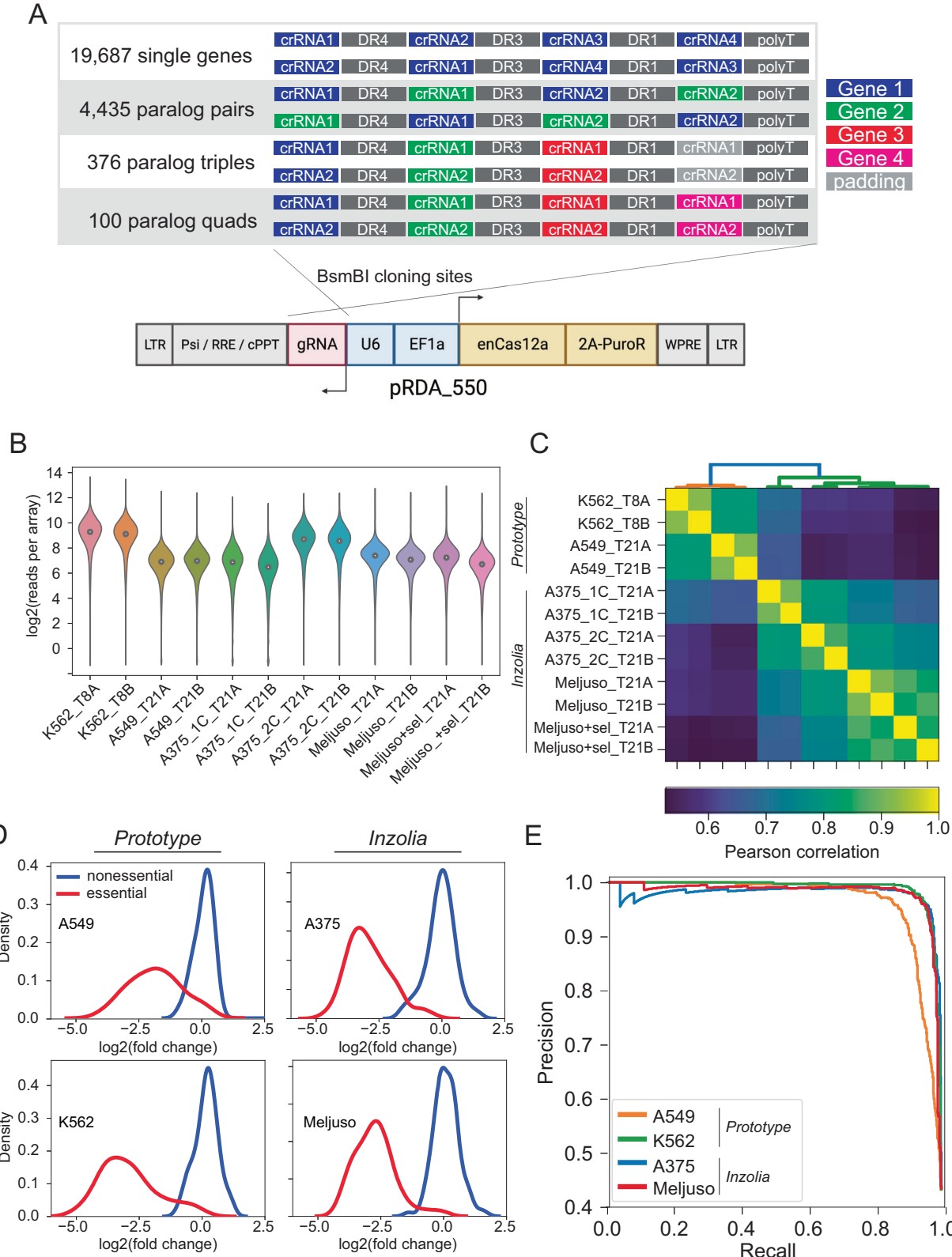

**Fig. 3 | In4mer platform for whole-genome screening. A** Inzolia human whole-genome library targets single genes and paralog pairs, triples, and quads with arrays of 4 Cas12a gRNA. Each gene or gene family is targeted by two arrays encoding the same gRNA in different order. Commercially synthesized oligo pools are cloned into the one-component pRDA_550 lentiviral vector; schematic created in Biorender. **B–E** Screening in K-562 CML cells (prototype library), A549 lung cancer cells (prototype library) and A375, MELJUSO melanoma cancer cells (Inzolia library). **B** Read counts from the plasmid and experimental timepoints after lentiviral transduction. **C** Correlation of sample read counts. Endpoint replicates are highly correlated. **D** Fold change distributions of arrays targeting reference essential (red) and non-essential genes (blue) in four cell lines. **E** Precision/recall analysis from ranked mean fold change of arrays targeting each gene, calculated against reference essential and non-essential genes.

library coverage, 8–10 doublings). Array amplicons were sequenced using single-end 150-base Illumina sequencing. Quality control metrics met expectations (Fig. 3B–E): the library was well-sampled in each replicate (Fig. 3B), and the abundance distributions of endpoint replicates were highly correlated (Fig. 3C). Fold changes showed increasing correlation when comparing clones targeting the same individual gene or paralog family within one replicate ($n = 22k$ targets, $r = 0.78$); all clones between two technical replicates derived from the same transduction ($n = 44k$ arrays, $r = 0.86$); and the mean of clones targeting the same gene across technical replicates ($n = 22k$ targets, $r = 0.92$; Supplementary Fig. 6).

Building on the success of the prototype library, we designed a second human genome library with several modifications. The updated library targets roughly twice as many paralogs (4435 pairs, 376 triples, and 100 quads; Fig. 3A, Supplementary Fig. 5, and Supplementary Data 5), includes non-targeting arrays to facilitate the estimation of fitness effects arising from multiplex locus-nonspecific DNA double strand breaks, and other minor technical changes such as using non-targeting sequences to extend 3mer paralog constructs into 4mer guide arrays, instead of random selection of non-essential guides as used in the prototype. This library, which we call Inzolia, contains ~49k unique arrays (Fig. 3A), and was cloned into both the pRDA_550 one-component vector and pRDA_052 guide-only expression vector for two-component CRISPR/Cas12a systems.

We screened the Inzolia library in MELJUSO melanoma cells with the two-component (split-vector) library and A375 melanoma cells with both the one- and two-component systems. Both cell lines effectively identified essential and non-essential genes (Fig. 3D,E) and the screens showed results consistent with previous Cas12a screens using the Humagne library and comparable to other CRISPR/Cas9 whole genome screens (Supplementary Fig. 7). Further, the one-component (pRDA_550) and two-component (pRDA_174 + pRDA_052) libraries yielded equivalent results (Supplementary Fig. 7).

Our prototype and Inzolia whole-genome libraries target small paralog families as well as single genes. To evaluate paralog genetic interactions, we used the multiplicative model to calculate the expected fitness of pairwise knockouts by summing the log fold change of the single gene knockouts (Fig. 1C). We then compared the observed mean fold change of guide arrays targeting gene pairs with the expected fold change under the multiplicative null model to calculate a dLFC that represents the magnitude of the genetic interaction (Fig. 4A). Gene pairs with strongly negative dLFC are highly concordant with the gold standard paralog synthetic lethals described above. The Inzolia library targets 24 of the 26 gene pairs that are hits in >1 of the previously published screens, and 12 of the 13 candidate gold standard paralog synthetic lethals with paralog scores ≥ 0.25. Of those 12, 9 have dLFC < −1 in MELJUSO cells, for an estimated sensitivity of 75% (Fig. 4B). Moreover, all 12 pairs (100%) with high paralog score are essential, regardless of synthetic lethality, as are 10 of 12 pairs (83%) with lower paralog scores (Fig. 4C), consistent with either synthetic lethality or one paralog being essential. Many other paralogs show genetic interactions as strong as these positive controls (see Fig. 4D for selected examples), with sequence similarity between paralogs being a strong predictor of GI (Fig. 4E), in keeping with prior observation[18].

The ability to recapitulate known biology is an important control for new technologies, with the MAP kinase pathway a frequently used case study in paralog buffering[22,24]. In K-562 cells, the *BCR-ABL* fusion oncogene activates the STAT and MAP kinase pathways, and we classify *ABL1*, *STAT5B*, and the *GRB2/SOS1/GAB2/SHC* signal transduction module as essential genes (Fig. 4F). None of the three *RAS* genes are individually essential, but the *KRAS-NRAS* pair shows a strong synthetic lethality. Neither *KRAS-HRAS* nor *HRAS-NRAS* paralogs show genetic interaction, but the three-way *HRAS-KRAS-NRAS* clones also show strong essentiality, almost certainly due to the *KRAS-NRAS* interaction.

In an arrayed validation screen, increased cell death after joint *KRAS-NRAS* knockout confirms this observation (Fig. 4G).

Beyond the RAS genes, the rest of the MAP kinase pathway also shows the expected gene essentiality profile in K-562 cells (Fig. 4H). *RAF1* is strongly essential, and while *BRAF* is slightly below our hit threshold, the *BRAF-RAF1* pairwise knockout is consistent with independent additive phenotype. The third member of the paralog family, *ARAF*, is non-essential singly or in combination with the other *RAF* paralogs and has not been shown to operate in this pathway. The MEK kinases, *MAP2K1/MAP2K2*, show greater fold change from pairwise loss than from either individually, though below our strict threshold for synthetic lethality. The ERK kinases, *MAPK1/MAPK3*, show strong preferential reliance on *MAPK1*, also consistent with DepMap data for K-562 cells.

Likewise, the other three cell lines we screened also show oncogene-driven essentiality and GI in the MAPK pathway (Fig. 4H). *KRAS* and *BRAF* essentiality in A549 and A375 cells, respectively, are consistent with driver mutations in those genes, though *NRAS* is not detected in *NRAS*-driven MELJUSO cells. A375 melanoma cells show isoform dependency on *MAPK1/MAP2K1*, also observed in DepMap screens in *BRAF*-driven melanoma cells. MELJUSO cells show interaction between MEK genes and both MELJUSO and A549 show GI between ERK genes. Data for all screens can be found in Supplementary Data 6 (read counts) and Supplementary Data 7 (log fold change).

Inzolia screens offer suggestions of polygenic (>2 gene) interactions as well. The library targets 476 paralog triples and quads, and several of these show indications of higher order synthetic lethality (Supplementary Fig. 8). We observe an intriguing interaction between *HSPA4* family of Hsp70-related chaperones, where *HSPA4* shows moderate phenotype when knocked out singly or in a pair with either family member, and severe phenotype when all three are targeted. This phenotype is highly variable across the four cell lines. Other candidate background-specific trigenic interactions include the *VDAC1/2/3* voltage-gated anion channel family and the *ME1/2/3* malic enzyme family, two of which were previously shown to be synthetic lethal in pancreatic cancer under nutrient limiting conditions[33]. Overall, however, distinguishing three-way synthetic lethal interactions from their composite pairs remains challenging. Even 80% single knockout efficiency can translate into $(0.8)^3 = ~50\%$ triple knockout efficiency, where cells with incomplete editing can mask severe triple knockout phenotypes.

Likewise, interactions between essential genes are also a challenge to interpret. Both the core proteasome and the Chaperonin-Containing TCP1 (CCT) complex are composed of several weakly related proteins, which we target with three four-way constructs and numerous two-way constructs. Since both the proteasome and the CCT complex are universally essential to proliferating cells, knockdown of single subunits induces a severe fitness phenotype. Knockout of these genes in pairs or quads yields no additional phenotype, resulting in what could be seen as masking/positive genetic interactions in all four cell lines (Supplementary Fig. 8). However, when the expected double knockout fitness exceeds the dynamic range of the assay – e.g. when the sum of two single knockout log fold changes is more severe than any observed fold change in the screen – a more conservative approach is to consider these pairs to be untestable rather than positive interactions.

Discovering genetic modifiers of drug activity is a common goal of genetic screens. To demonstrate the performance of in4mer constructs in discovering chemogenetic interactions, we screened MELJUSO cells in the presence of low-dose MEK inhibitor selumetinib. DrugZ analysis[9] of single gene targets (Fig. 5A and Supplementary Data 8) shows that genetic perturbation of the MAP kinase pathway sensitizes cells to the drug (Fig. 5A, B). While gene set enrichment analysis[34,35] identifies subunits of the mitochondrial ribosome,

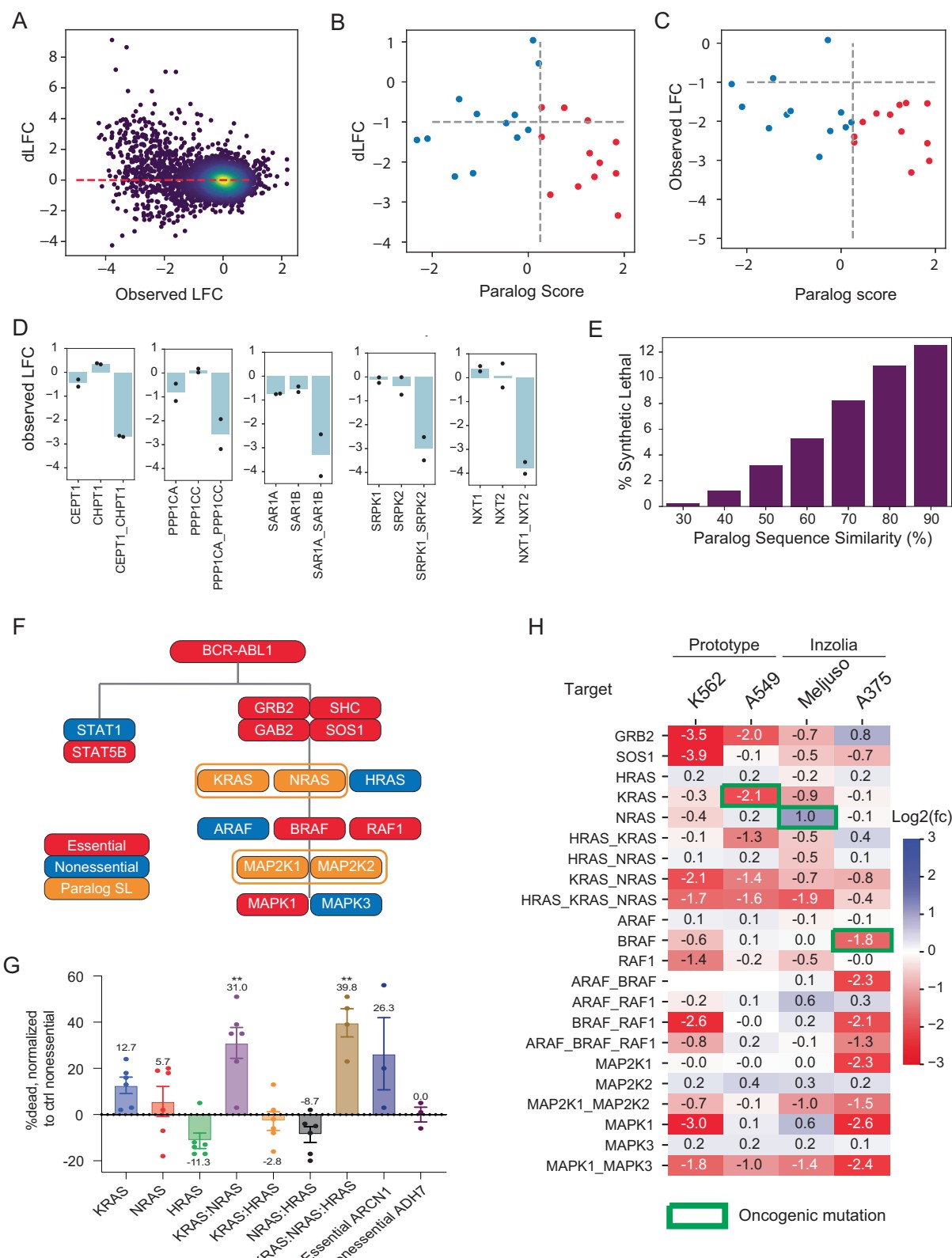

components of the peroxisome, and elements of the Hippo pathway (Fig. 5B and Supplementary Data 9) as suppressor genes, only two Hippo pathway genes achieved high Z-scores (Fig. 5A). DrugZ analysis of pairwise paralog knockouts yielded hits generally consistent with single gene knockout; that is, most paralog knockouts give DrugZ scores consistent with the most extreme single gene knockout (Fig. 5C). In some cases, however, combinatorial perturbation of

paralogs gave rise to synergistic effects, indicating genetic buffering of chemogenetic interaction. Notably, of the five paralog combinatorial knockouts with Z-scores > 5 and significantly greater effect than their singletons, three encode redundant elements of the Hippo pathway, including *STK3/STK4, LATS1/LATS2,* and *MOB1A/MOB1B* (Fig. 5C, D).

In total, the Inzolia library includes ~50k unique guide arrays, with ~40k targeting single genes and 9822 arrays targeting paralog doubles,

**Fig. 4 | Paralog synthetic lethality with Inzolia. A** Fold change vs. dLFC for >4000 paralog families in MELJUSO cells. **B** dLFC vs. Paralog Score from meta-analysis of published paralog screens. Red, Paralog Score > 0.25. Blue, Paralog Score <0.25. Of 12 paralogs with score > 0.25, 9 show dLFC < −1 in MELJUSO cells. **C** Fold change vs. paralog score in MELJUSO cells, color as in **B**. Most scored paralogs are essential, regardless of synthetic lethality. **D** Selected synthetic lethals in MELJUSO cells showing single and double knockout fitness phenotype. Bar chart, mean fold change. Points indicate fold change of single array of gRNA (mean of 2 replicates). **E** Fraction of synthetic lethal paralogs by amino acid sequence similarity in MEL-JUSO cells. **F** Pathway activation by *BCR-ABL1* fusion in K-562 cells. Red, essential gene in in4mer screen; blue, non-essential; orange, synthetic lethal paralog pair.

**G** Fraction of dead cells, normalized to controls, for single, double, and triple knockouts of RAS genes in K-562. Two clones were used for each gene/gene combination, and three technical replicates were maintained for each clone, *n* = 6 for each condition/group. *KRAS-NRAS* double and *KRAS-NRAS-HRAS* triple knockout show significantly increased cell death compared to negative control (**\*P* < 0.01, one-way ANOVA). Data represented as mean ± SEM. *ARCN1*, control essential gene. *ADH7*, control non-essential gene. **H** Single, double, and triple knockout phenotype of RTK/MAP kinase pathway genes in all four cell lines. White, target not in library. Green outline, known oncogenic mutation. Source data are provided as a Source Data file.

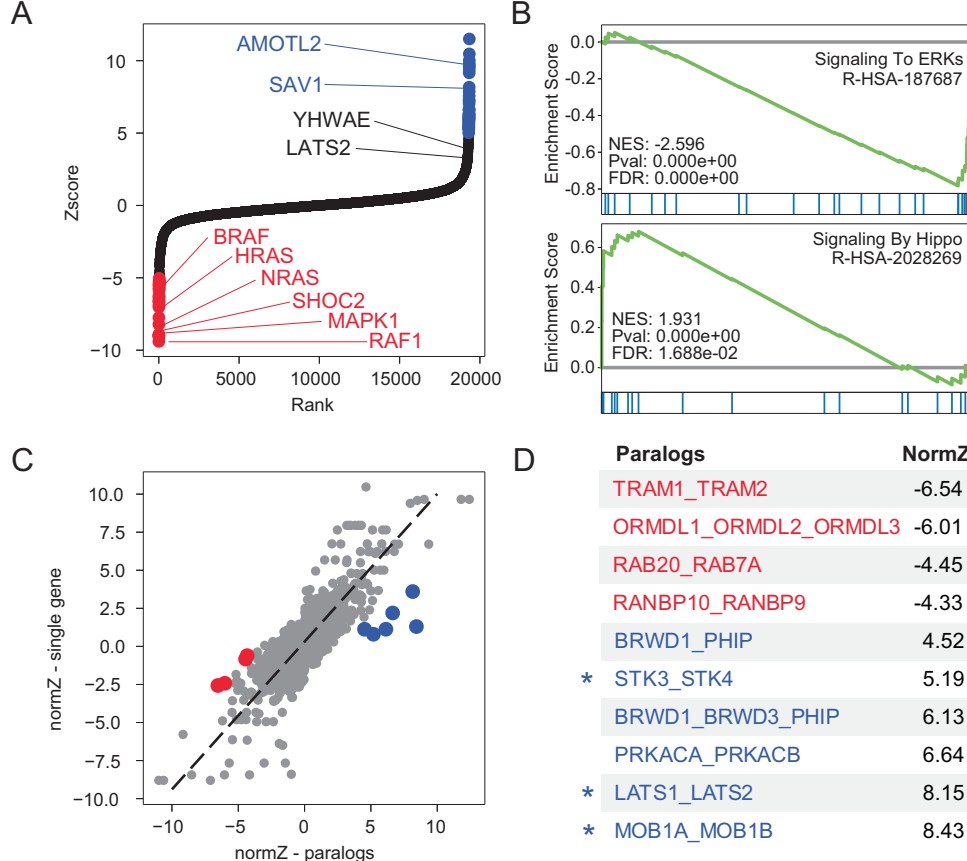

**Fig. 5 | Synthetic chemogenetic interactions.** MELJUSO cells were cultured in the presence of MEK inhibitor selumetinib and screened for chemogenetic interactions (**A**) DrugZ scores of single gene knockouts. Selected genes in the MAPK and Hippo pathways highlighted. Red, synergistic; blue, suppressor. **B** Selected GSEA results and significance tests (one-sided weighted Kolmogorov–Smirnov statistic) for gene sets conferring sensitivity (ERK signaling) or resistance (Hippo signaling) to MEKi.

**C** Comparing DrugZ scores of paralogs (*x*-axis) vs. the most extreme Z score of the single gene knockout (*y*-axis) shows that most pairwise perturbagens yield similar phenotype as singletons. Outliers in red (synergistic) or blue (suppressor). **D** Synergistic and suppressor paralog knockouts from **C**. Asterisk indicates functional buffering in the Hippo pathway, masking phenotype in monogenic knockout screens.

triples, and quads. Inzolia is therefore on par with latest-generation genome-scale CRISPR/Cas knockout libraries[27,36–39] (Fig. 6), and is unique among such libraries in including thousands of reagents targeting paralogs. Moreover, the efficiency gain realized by having two guides targeting each of two genes in a paralog pair makes detection of genetic interactions tractable with only six reagents per gene pair, a fivefold improvement over the prior state of the art (Fig. 6).

## Discussion

The rapid ascendancy of CRISPR-mediated genetic perturbation technologies over RNA interference methods was driven by major advances in assay sensitivity and specificity, with the absence of established gold standards arguably contributing to the shortcomings of RNAi-based studies of mammalian gene function[40]. We and others

have created widely used reference sets of essential and non-essential genes for use in quality control of monogenic loss of fitness screens[2,39,41]. As CRISPR perturbation technology has advanced into genetic interactions, it has become clear that a similar gold standard for synthetic lethals is needed[31]. Our meta-analysis of published screens for paralog synthetic lethals in human cells shows wide divergence in the paralogs assayed by each study and in the repeatability of each screen, as measured by the Jaccard coefficient of hits in different cell lines. We reasoned that paralogs that showed synthetic lethality within and across screening platforms are likely to be globally synthetic lethal, analogous to core essential genes, and the fact that 12 of our 13 candidate reference paralogs show more than 70% identity (and all are constitutively expressed) is consistent with this interpretation.

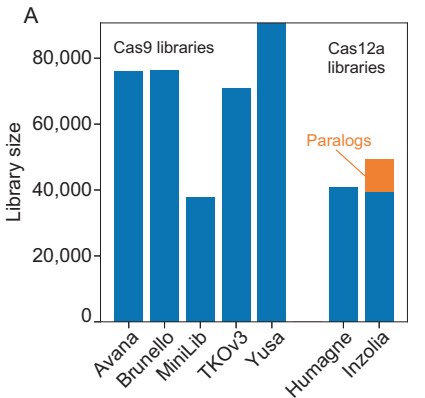

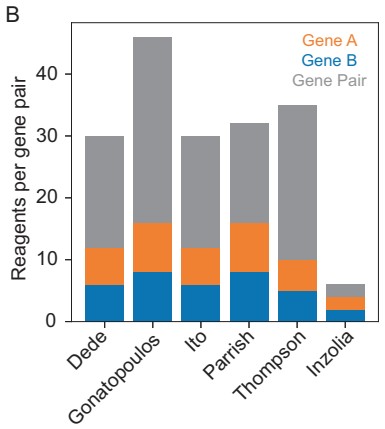

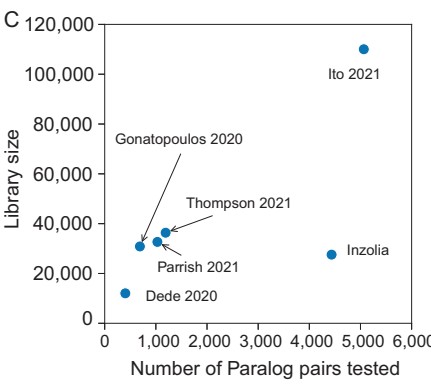

**Fig. 6 | Library size comparison. A** Representative Cas9 and Cas12a whole-genome libraries. Inzolia library targets 19k protein-coding genes and additionally includes 9822 guide arrays targeting paralog doubles, triples, and quads. **B** Five recent publications screening for genetic interactions between paralogs. Bar plot shows number of reagents per paralog pair tested, including single and double knockouts.

**C** Comparison of library efficiency. Number of paralog pairs tested vs. library size for recent publications. For this plot, Inzolia library only includes paralog doubles (4435), triples (376), quads (100), and plus corresponding single gene knockouts (8870).

Notably, the engineered Cas12a endonuclease, developed in Kleinstiver et al.[26] and deployed in combinatorial screens in DeWeirdt et al.[27] and paralog screens in Dede et al.[19] performed markedly better in terms of replicability. Based on this and our prior work with the CRISPR/Cas12a screens[27,28], we tested the limits of the Cas12a system expressing guide arrays from the Pol III U6 promoter in a custom one-component lentiviral vector, pRDA_550. For longer arrays of seven independent gRNA, we observed that position-specific loss of knock-out efficiency did not arise until after the fourth or fifth gRNA in the array. From this we developed the in4mer platform for arrays encoding four independent gRNA, each with an optimized spacer sequence from the CRISPick algorithm and with diverse but proven DR sequences to minimize the chance of recombination. By targeting single genes with four independent gRNA, we lower the odds that any single guide fails to induce the desired phenotype, extending the development of the Humagne library in work from DeWeirdt et al.[27]. As with the Humagne library, having multiple independent gRNA on each array reduces the total number of reagents required to induce reliable gene perturbation.

While Cas12 exhibits advantages in multiplexing, its success relies on selecting a suitable biological model and ensuring optimal gRNA efficiency. Similar to Cas9, a potential challenge in screening with Cas12 is the induced double-strand breaks, triggering a DNA damage response and subsequent cell cycle arrest. Previous studies have highlighted that the number of loci targeted by CRISPR, particularly those spanning chromosomes, can result in gene-independent fitness loss, potentially leading to a higher rate of false-positive identification of undesired cell-essential genes[42]. Although Berg et al.[43] has revealed the limited differences between one cut and four cuts resulting in H2AX changes, and most cancer cell lines exhibit tolerance to some extent of DNA damage response, the potential for high copy numbers and off-target effects induced additional breaks may compromise accuracy.

To construct our Inzolia genome-scale human library, we began with reagents targeting single protein-coding genes and added arrays targeting more than four thousand paralog pairs, triples, and quads, with the in4mer arrays encoding two guides targeting each of the two genes in a pair or one guide per gene in a triple or quad family. Inzolia screens show high (at least 75%) sensitivity to detect synthetic lethals with just two reagents targeting each single knockout and two reagents targeting the double knockout, and offer the potential for novel biology arising from three- and four-way paralog synthetic lethals. The Inzolia library is thus a smaller and more efficient whole-genome library that addresses one of the major gaps of monogenic perturbation libraries: functional buffering by paralogs.

Beyond the paralogs in the Inzolia library, the in4mer system is a highly customizable platform offering a significant advance in the study of genetic interactions. Compared to the five paralog synthetic lethal studies, with each using at least thirty constructs per gene pair tested, in4mer requires fivefold fewer reagents for the same assay. This improvement has major implications for the cost effectiveness in genetic interaction assays in mammalian cells, where the number of gene pairs and the diversity of cell/tumor lineages and genotypes combine to yield a vast search space. A fivefold reduction in experimental footprint could offer a correspondingly larger search space for the same effort, or the same search space across more backgrounds (e.g. cell lines) or environments (e.g. chemogenetic interactions) at nearly the same cost as a single screen with an equivalent combinatorial Cas9 system, with the Cas12a system yielding greater sensitivity and robustness. Moreover, custom library construction leverages the other key advantage of the Cas12a system: each library is constructed from a single ~200mer oligo pool and both cloning and amplicon sequencing are performed using essentially the same protocols as single-guide Cas9 screening, albeit with longer sequencing reads. With the in4mer system, a wider swath of the research community will be able to add targeted genetic interaction surveys to their experimental toolkits.

## Methods
### Paralog meta-analysis
To reanalyze the data from the 5 paralog screens, raw read counts were downloaded, and the same pipeline was applied to all of them. A pseudocount of 5 reads was added to each construct in each replicate, and total read counts were normalized to 500 reads per construct. Log2 fold change (LFC) for each guide at late time point was calculated relative to the plasmid sequence counts.

The data from each study (except Thompson) were divided into three groups; the constructs that target single genes paired with non-essential/non-targeting gRNAs (N) in the first position (gene_N), in the second position (N_gene) and constructs that target gene pairs (A_B). LFC values of each group were scaled individually so that the mode of each group was set to zero. Next, all three groups were merged in one table. Before dividing Ito's dataset into three groups, LFC values were scaled such that the mode of negative controls (non-essential_AAVS1) would be zero and also TRIM family was removed from this dataset to avoid false paralog pair discovery[13]. Since in Thompson's study there

was just one position for singleton constructs, LFC values were scaled so that the mode of negative controls (non-essential_Fluc) was set to zero. In the next step, LFC of each construct was calculated by the mean of LFC across different replicates.

To calculate genetic interaction, single gene mutant fitness (SMF) was calculated as the mean construct log fold change of gene-control constructs for each gene. The control was either non-essential genes or non-targeting gRNAs. For each gene pair, the expected double mutant fitness (DMF) of genes 1 and 2 was calculated as the sum of SMF of gene 1 and SMF of gene 2. The difference between expected and observed DMF, the mean LFC of all constructs targeting genes 1 and 2, was called dLFC.

Next step was calculating a modified Cohen's D between observed and expected distribution of LFC of gRNAs targeting genes. Expected distribution of gRNAs targeting a gene pair, was calculated using expected mean and expected standard deviation (std).

$$expected\ mean = \mu1 + \mu2 \tag{1}$$

$$expected\ stand\ deviat = \sqrt{(std1)^2 + (std2)^2} \tag{2}$$

$$S_{pooled} = \frac{\sqrt{(expected\ std)^2 + (observed\ std)^2}}{2} \tag{3}$$

$$Cohen's D = \frac{expected\ mean - observed\ mean}{S_{pooled}} \tag{4}$$

Where $\mu1$ = mean LFC of gene1 constructs, $\mu2$ = mean LFC of gene2 constructs, $std1$ = standard deviation of LFC of gene1 constructs, and $std2$ = standard deviation of LFC of gene2 constructs.

In each cell line, the paralog pairs with dLFC < −1 and Cohen's D > 0.8 were selected as hits. Cohen's D > 0.8 indicates large effect size between two groups, meaning that our expected and observed distribution of gRNAs are meaningfully separated. In total 388 paralog pairs were identified as hits across all the studies.

To identify the most consistent method in terms of hit identification, the Jaccard similarity coefficient of every pair of cell lines in each study was calculated by taking the ratio of intersection of hits over union of hits. For the studies that screened more than two cell lines, the final platform weight was the median of the calculated Jaccard coefficients of all pairs of cell lines.

$$J(A, B) = \frac{|A \cap B|}{|A \cup B|} = \frac{|A \cap B|}{|A| + |B| - |A \cap B|} \tag{5}$$

To score paralog pairs, each hit was scored based on the cell lines in which it was identified as a hit; cell lines were weighted based on the platform weight described above. We defined the "paralog score" as the sum of platform weights of cell lines in which the paralog pair was identified as a hit minus the sum of platform weights of cell lines in which the paralog pair was assayed but not identified as a hit (a "miss"). The distribution of scores is shown in Fig. 1. Gene pairs with paralog score > 0.25 and were identified as a hit in two or more studies were listed as candidate gold standard paralog synthetic lethals.

## One-component CRISPR/enCas12a vector

To construct an all-in-one vector for expression of both Cas12a and a guide array, we first swapped in puromycin resistance in place of blasticidin resistance from pRDA_174 (Addgene #136476). We then tested four locations for the insertion of a U6-guide expression cassette; notably this cassette needs to be oriented in the opposite direction of the primary lentiviral transcript to prevent Cas12a-mediated processing during viral packaging in 293 T cells. The

construct with the best-performing location, between the cPPT and the EF-1α promoter, was designed pRDA_550 (Addgene #203398). Synthesis of DNA and custom cloning was performed by Genscript.

## 7mer library production

An oligonucleotide pool consisting of 7 Essential and 7 Non-Essential gene crRNAs with their nearby DR, BsmBI recognition as well as overhang sequence was synthesized by Integrated DNA Technologies. The pool was amplified by asymmetric PCR followed by being assembled into PRDA_550 vector to acquire the designed library through NEBridge® Golden Gate Assembly Kit (BsmBI-v2) (New England Biolabs). The assembled product was transformed into NEB® Stable Competent E. coli (High Efficiency) cells and the plasmid DNA was purified using the PureLink™ Plasmid Purification Kit (Invitrogen). Three oligonucleotide pools were cloned separately and pooled together to acquire the final 7mer library. The library was sequenced to confirm uniform and complete library representation.

## Paralog selection for In4mer/Inzolia

Human paralogs and percent identity data were imported from BioMart, which reports both AB and BA percent identity (these can differ if the two genes encode proteins of different lengths) Mean percent identity ((AB + BA)/2)and delta percent identity (|AB−BA|) between paralogs were then calculated, and for the prototype library, paralogs with mean percent identity between 30% and 99% and delta percent identity <10% were selected (Supplementary Fig. 5). Next, CCLE expression data was downloaded, and the mean and standard deviation of expression across all CCLE samples was calculated for each gene. Paralogs where both genes had mean expression > 2 and stdev < 1.5 were selected (i.e. constitutively expressed genes).

Finally, to identify and include paralog families of size > 2, we applied a "difference from top paralog" filter. For each gene A in the pool, we identified its top paralog B by max sequence identity. Then for each other candidate paralog C, we calculated the drop in sequence identity, AB−AC (see distribution of drop % in Supplementary Fig. 5). For the prototype library, we defined A,B,C as being in the same family if AB−AC < 10%.

For the final Inzolia library, we relaxed several of these filters. The delta percent identity filter and the expression variance filter were removed entirely, and the difference from top paralog filter was expanded to 20%. The mean expression filter was retained. These three filtering steps resulted in a total of 4435 paralog pairs included in the Inzolia pool library.

## In4mer prototype library production

Oligonucleotide pools consisting of designed four-plex guide arrays were synthesized by Twist Bioscience. The prototype pool consists of 43,972 arrays targeting 19,687 single genes, 2082 paralog pairs, 167 paralog triples, and 48 paralog quads.

5'-*AATGATACGGCGACCACCGA***cgtctcgA-GAT**nnnnnnnnnnnnnnnnnnnnnnn<u>TAATTTCTACTATTGTA-GAT</u>nnnnnnnnnnnnnnnnnnnnnnn<u>AAATTTCTACTCTAGTA-GAT</u>nnnnnnnnnnnnnnnnnnnnnnn<u>TAATTTCTACTGTCGTA-GAT</u>nnnnnnnnnnnnnnnnnnnnnnnn<u>TTTTTT</u>**GAAT**gga-gacg*ATCTCGTATGCCGTCTTCTGCTTG*-3'.

Italic: primer sequence Bold: BsmBI restriction sequence. Overhang in CAPS. nnnnn: guide sequence Underlined: DR sequence.

The pool of guide arrays was PCR amplified using KAPA HiFi 2X HotStart ReadyMix (Roche) using 20 ng of starting template per 25 µL reaction (primers are listed in Supplementary Data 10) and the following conditions: denaturation at 95 °C for 3 min, followed by 12 cycles of 20 s at 98 °C, 30 s at 60 °C, and 30 s at 72 °C, followed by a final extension of 1 min at 72 °C. The resulting amplicon was purified by the Monarch PCR & DNA Cleanup Kit (New England Biolabs) and cloned into the pRDA-550 vector by NEBridge® Golden Gate Assembly

Kit (BsmBI-v2) The product from assembly reaction was purified and electroporated into Endura Electrocompetent cells (Lucigen). Transformed bacteria were diluted 1:100 in 2xYT medium containing 100 µg/mL carbenicillin (Sigma) and grown at 30 °C for 16 h. The plasmid DNA was extracted by PureLink™ Plasmid Purification Kit (Invitrogen). The library was sequenced to confirm uniform and complete library representation. The library was prepared in MD Anderson Cancer Center.

## Inzolia library production

The final Inzolia pool consists of arrays targeting 19,687 single genes, 4435 paralog pairs, 376 paralog triples, and 100 paralog quads, plus 20 arrays targeting EGFP, 500 targeting intergenic loci, and 50 encoding non-targeting guides. Each array in the oligonucleotide pools is constructed as follows:

*5′-AGGCACTTGCTCGTACGACG***cgtctcgA-GAT**nnnnnnnnnnnnnnnnnnnnnnnTAATTTCTACTATTGTA-GATnnnnnnnnnnnnnnnnnnnnnnnAAATTTCTACTCTAGTA-GATnnnnnnnnnnnnnnnnnnnnnnnTAATTTCTACTGTCGTA-GATnnnnnnnnnnnnnnnnnnnnnnnTTTTTT**GAAT**gga-gacg*TTAAGGTGCCGGGCCCACAT-3′*.

Italic: primer sequence Bold: BsmBI restriction sequence. Overhang in CAPS. nnnnn: guide sequence Underlined: DR sequence.

The pool of guide arrays was PCR amplified using NEBNext® High-Fidelity 2X PCR Master Mix (NEB) using 196 ng of starting template per 50 µL reaction (primers are listed in Supplementary Data 10) and the following conditions: denaturation at 98 °C for 1 min, followed by 7 cycles of 30 s at 98 °C, 30 s at 53 °C, and 30 s at 72 °C, followed by a final extension of 5 min at 72 °C. The resulting amplicon was purified by the Qiaquick PCR Purification Kit (Qiagen) and cloned into the pRDA-550 and pRDA-052 via Golden Gate cloning with Esp3I (Fisher Scientific) and T7 ligase (Epizyme). The assembly product was purified by isopropanol precipitation, electroporated into Stbl4 electrocompetent cells (Life Technologies) and grown at 37 °C for 16 h on agar with 100 ug/mL carbenicillin. Colonies were scraped and plasmid DNA (pDNA) was extracted via HiSpeed Plasmid Maxi (Qiagen). The library was sequenced to confirm uniform and complete library representation. The library was prepared in Broad institute.

## Cell culture

K-562 and A549 cells were a gift from Tim Heffernan. A375 and MEL-JUSO were obtained from the Cancer Cell Line Encyclopedia. Cell line identities were confirmed by STR fingerprinting by M.D. Anderson Cancer Center's Cytogenetic and Cell Authentication Core. All cell lines were routinely tested for mycoplasma contamination using cells cultured in non-antibiotic medium (PlasmoTest Mycoplasma Detection Assay, InvivoGen).

All cell lines were grown at 37 °C in humidified incubators at 5.0% $CO_2$ and passaged to maintain exponential growth. For each cell line, the following medium and concentration of polybrene (EMD Millipore) and puromycin (Gibco) were used:

K-562: RPMI + 10% FBS, 8 µg/mL, 2 µg/mL
A549: DMEM + 10%FBS, 8 µg/mL, 2 µg/mL
A375: RPMI + 10% FBS, 1 µg/mL, 1 µg/mL
MELJUSO: RPMI + 10% FBS, 4 µg/mL, 1 µg/mL.

## Cas12a screens

Lentivirus was produced by the University of Michigan Vector Core (prototype) or the Broad GPP (Inzolia). Virus stocks were not titered in advance. Transduction of the cells was performed at 1X concentration of virus with corresponding polybrene. Non-transduced cells were eliminated via selection puromycin dihydrochloride. The selection was maintained until all non-transduced control cells reached 0% viability. Once selection with puromycin was complete, surviving cells were pooled and 500x coverage cells were harvested for a T0 sample. After T0, cells were harvested at 500X coverage on corresponding days. The prototype In4mer screens were performed in MD Anderson Cancer Center. The Inzolia screens were performed in Broad Institute.

## Prototype In4mer library genomic DNA preparation and sequencing

Genomic DNA (gDNA) was extracted using the Mag-Bind® Blood & Tissue DNA HDQ 96 Kit (Omega Bio-tek) and quantified by the Qubit™ dsDNA Quantification Assay Kits (ThermoFisher). Illumina-compatible guide array amplicons were generated by amplification of the gDNA in a one-step PCR. Indexed PCR primers were synthesized by Integrated DNA Technologies using the standard 8nt indexes from Illumina (D501-D508 and D701-D712) (Supplementary Data 10).

At least ~200X coverage gDNA per replicate across multiple reactions were amplified. Each gDNA sample was first divided into multiple 50 µL reactions with most 2.5ug gDNA per reaction. Each reaction contained 1ul each primer (10 µM), 1 µL 50X dNTPs, 5% DMSO, 5 µL 10X Titanium Taq Buffer, and 1 µL 10X Titanium Taq DNA Polymerase (Takara). The PCR conditions were: denaturation at 95 °C for 60 s, followed by 25 cycles of 30 s at 95 °C and 1 min at 68 °C, followed by a final extension at 68 °C for 3 min. After the PCR, all reactions from the same sample were pooled and then purified by E-Gel™ SizeSelect™ II Agarose Gels, 2% (ThermoFisher). Purified amplicons were quantified by Qubit™ dsDNA Quantification Assay Kits (ThermoFisher) and validated by D1000 ScreenTape Assay for TapeStation Systems (Agilent) (360 bp for in4mer, 501 bp for 7Mer). Purified amplicons were then pooled (with 30% customized random library to increase the diversity) and sequencing was performed by NextSeq 500 sequencing platform (Illumina) with custom primers (Integrated DNA Technologies) (Supplementary Data 10). The In4mer library was sequenced by read format of 151-8-8, single-end and the 7Mer library was sequenced by read format of 151-8-8-151, paired-end.

## Inzolia genomic DNA preparation and sequencing

Genomic DNA (gDNA) was extracted using the Mag-Bind® Blood & Tissue DNA HDQ 96 Kit (Omega Bio-tek) and quantified by the Qubit™ dsDNA Quantification Assay Kits (ThermoFisher). Illumina-compatible guide array amplicons were generated by amplification of the gDNA in a one-step PCR. Indexed PCR primers were synthesized by Integrated DNA Technologies using the standard 8nt indexes from Illumina (D501-D508 and D701-D712). The sequences for the primer sets were listed in Supplementary Data 10.

At least ~200X coverage gDNA per replicate across multiple reactions were amplified. Each gDNA sample was first divided into multiple 100 µL reactions with most 10 µg gDNA per reaction. Each reaction contained 0.5 µL forward primer (100 µM), 10uL reverse primer (5 uM) 8 µL dNTPs, 5 µL DMSO, 10 µL 10X Titanium Taq Buffer, and 1.5 µL Titanium Taq DNA Polymerase (Takara). The PCR conditions were: An initial denaturation at 95 °C for 60 s, followed by 28 cycles of 30 s at 94 °C, 30 s at 52 °C, and 30 s at 72 °C followed by a final extension at 72 °C for 10 min. After the PCR, all reactions from the same sample were pooled and purified with Agencourt AMPure XP SPRI beads according to the manufacturer protocol (Beckman Coulter). Purified amplicons were quantified by Qubit™ dsDNA Quantification Assay Kits (ThermoFisher) and sequenced on a HiSeq2500 with a Rapid Run (200 cycle) kit (Illumina).

## 7mer screen data analysis

Reads for each reagent were counted using only exact matches to the entire 281 nucleotide 7mer sequence, excluding the leading DR (7 23mer spacer sequences + 6 20mer DR sequences). Fold changes were calculated relative to the mean of the T0 samples, and averaged across replicates. For each sample (T7/14/21), fold changes were normalized by subtracting the mean fold change of arrays with 7 nonessentials; i.e. setting no-essentials guides to zero.

We expected that the selected essential genes would not show any pairwise or higher order interactions, and thus should be governed by the multiplicative model of genetic interaction. To evaluate this model, we fit a regression model:

$$y \sim A\beta \qquad (6)$$

where A is a binary matrix of 7mer guide arrays (rows, $k = 384$) by positions (columns, $n = 7$), with $A_{i,j} = 1$ if guide array i targets an essential gene at position $j$ and 0 if not. $y$ is the vector of normalized observed fold changes, and the n-length vector $\beta$ coefficients represent the single gene knockout phenotype learned from the model. We filtered this construct for reagents that encoded two or fewer essential genes ($k = 87$ rows). After linear fit, we compared the predicted zero, one, and two gene knockout fitness profiles (by summing the $\beta$ coefficients for each gene) to the mean observed knockout fitness. $R^2$ values for each pool ranged from 0.78 to 0.91, and the overall quality of the linear fit supports the multiplicative model for non-interacting genes as assayed by combinatorial CRISPR knockouts of up to two genes. An accurate null model for noninteraction is critical for detecting and classifying deviations from this model that reflect positive or negative genetic interactions.

### In4mer/Inzolia screen data analysis

In4mer library sequencing reads were mapped to the library using only perfect matches. BAGEL2 was used to normalize sample level read counts and to calculate fold changes relative to the T0 reference using the BAGEL2.py *fc* option with default parameters[44]. Essential and non-essential genes were defined using the Hart reference sets from refs. 39,41. Since the library targets both individual genes and specific gene sets (paralogs), we calculated the average gene/gene set (hereafter 'gene') log fold change as the mean of the clone-level fold changes across two replicates. All fold changes are calculated in log2 space. Cohen's D statistics were calculated in Python as described in Paralog meta-analysis above. Data for recall-precision curves were calculated using BAGEL2. We set an arbitrary threshold of fc < −1 for essential genes.

For genetic interaction analysis, the expected fold change was calculated as the sum of the gene-level fold changes for each individual gene in the gene set. Expected fc was subtracted from observed fc to calculate delta log fold change, dLFC, where negative dLFC indicates synthetic/synergistic interactions with more severe negative phenotype, and positive dLFC indicates positive/suppressor/masking interactions with less severe negative or more positive phenotype than expected. We set an arbitrary threshold of dLFC < −1 for synthetic lethality, and >+1 for masking/suppressor interactions.

### RAS synthetic lethal validation

An arrayed knockout apoptosis assay approach was adopted to validate *RAS* synthetic lethality in K-562. Two guides were selected for each of the three *RAS* genes, and two clones were designed for each target/gene combination. Guide RNAs were selected through CRISPick and gblocks (same construct as Inzolia library) were synthesized by Integrated DNA Technologies. The arrays were individually cloned into the pRDA_550 backbone and plasmids were validated by Sanger sequencing. The plasmids were then individually transfected to K-562 cells via the Neon Transfection System (Invitrogen). Each group was transfected with 2 µg of DNA per $2 \times 10^6$ cells, using the recommended setting for K-562 electroporation with one pulse at 1000 v, 50 ms. Non-transfected cells were eliminated through puromycin selection, which was maintained until non-transfected control cells reached 0% viability. Triplicate wells were maintained after selection until the end of the experiment. Cell viability, total cell numbers, live cell size and dead cell size data were collected through reading Trypan Blue (Gibco) stained cells via Countess II FL (Thermo Fisher) at each passage until 9 days after puromycin selection, in line with Inzolia screen end point of 8 days in K-562 cells. Percent dead cells were normalized to negative control and one-way ANOVA was conducted to compare experimental groups against the negative control for statistical significance.

### Reporting summary

Further information on research design is available in the Nature Portfolio Reporting Summary linked to this article.

## Data availability

The Inzolia library is available on Addgene as catalog #209551 and #209552. All data generated in this study have been deposited in Figshare [https://doi.org/10.6084/m9.figshare.24243832.v1][45]. Source data are provided with this paper.

## Code availability

Meta-analysis of 5 dual-knockout CRISPR studies, Inzolia library design, In4mer data analysis were conducted in Python 3.8.10, using the packages Pandas 2.1.4, SciPy 1.11.4, NumPy 1.26.2, glob and statistics. Seaborn 0.13.0 and Matplotlib 2.8.2 were used to generate plots and upsetplot 0.9.0 package was used for upset plot generation. All code for this study is available at Figshare [https://doi.org/10.6084/m9.figshare.24243832.v1][45].

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

## Acknowledgements
N.E.A., L.L.W., X.M., M.C. and T.H. were supported by NIGMS grant R35GM130119, NCI grant U01CA275886, and CPRIT grants RP210173 and RP210073. N.E.A. and S.H.W. are supported by National Institutes of Health grant R01GM139980. T.H. is a CPRIT Scholar in Cancer Research and an Andrew Sabin Family Fellow. C.L. is an Odyssey Fellow and is supported by the Odyssey Program and Odyssey Expansion Fund at MD Anderson. This work was additionally supported by the NCI Cancer Center Support Grant P30CA16672 (TH) and NCI Cancer Moonshot Initiative U01CA250565 (J.G.D.).

## Author contributions
N.E.A performed paralog meta-analysis. N.E.A., A.K.S., S.H.W. and J.G.D. designed, performed, and analyzed guide efficiency profiling experiments. C.L. performed 7mer screens and bioinformatic analysis. A.K.S. and J.G.D. designed the pRDA_550 plasmid. C.L, L.L.W. and X.M. performed in4mer prototype screens; and R.S. and A.K.S. performed in4mer Inzolia screens. N.E.A., C.L., M.C., R.S. and X.M. performed in4mer bioinformatic analysis. X.M. performed RAS synthetic lethal validation. S.H.W., J.G.D. and T.H. supervised the research. N.E.A., C.L., X.M. and T.H. drafted the manuscript and all authors edited it.

## Competing interests
J.G.D. consults for Microsoft Research, Abata Therapeutics, Servier, Maze Therapeutics, BioNTech, Sangamo, and Pfizer. J.G.D. consults for and has equity in Tango Therapeutics. J.G.D. serves as a paid scientific advisor to the Laboratory for Genomics Research, funded in part by GlaxoSmithKline. J.G.D. receives funding support from the Functional Genomics Consortium: Abbvie, Bristol Myers Squibb, Janssen, Merck, and Vir Biotechnology. J.G.D.'s interests were reviewed and are managed by the Broad Institute in accordance with its conflict of interest policies. Other authors don't claim competing interests.
