## [Peer Review File · Nature Communications]

Efficient gene knockout and genetic interaction screening using the in4mer CRISPR/Cas12a multiplex knockout platformReviewers' Comments:

Reviewer #1:

Remarks to the Author:

In Anvar et al., the authors compare their combinatorial screening platform, which uses Cas12a, to other previous screens which used Cas9. They also describe a new library generated using the Cas12a system. The IN4MER platform is the state-of-the-art – that said I have a number of specific comments on this paper.

1. This paper doesn't have any supplementary tables or links to Github or FigShare. As I read the paper, I was looking for tables that had the gRNA sequences, the hits from the various screens performed, and the count files but I was unable to locate these. There are other elements of the analysis that I would really like to know – for example the identity of the 388 gene pairs that were scored as hits across all five multiplex perturbation platforms (Figure 1E). I would also like to know the hits called in each of the different screens. Because I don't have this information it makes it difficult for me to understand if the screen hits make biological sense. It would also be interesting to know which paralog families are SL. I can't find any of this information in the manuscript.
2. In comparing screens, the authors don't appear to consider differences in experimental design. For example, each of the different screens have been sequenced to a different level of coverage, which might influence the ability to detect "hits". The screens were also performed at different timepoints. Collectively, these variables reflect aspects of screen set-up rather than the performance of the different Cas systems/vectors and the role they may play are not discussed.
3. There are lots of messages in this paper. That the IN4MER platform is "better", that Cas12a screens when performed in the way that the authors describe can be used to identify and explore paralogs as pairs and additional paralog gene sets etc. It feels to me that the authors are holding a biological story back or have plans to publish the screen results elsewhere. If that's the plan, then the authors should just say that.
4. A library like the one described in the paper with multiple identical repeats might be expected to undergo recombination. Have the authors systematically examined this?

Notwithstanding the comments above the paper describes an advance that will be useful to the field.

Reviewer #2:

Remarks to the Author:

Anvar et al. generated a Cas12a crRNA platform, termed In4mer, where 4 crRNAs can be cloned into each vector molecule and tested their genetic interactions. Using this platform, they generated a new Cas12a library, Inzolia, targeting singles, pairs, trios and quads with approximately 50k clones, substantially wider targets with a smaller size. Screens with this library identified expected genetic interactions in fitness and under drug treatment.

The 4-crRNA platform is an extension of the previous discovery by the authors (Nat Biotechnol 2021), but has made wider applications possible. In particular, assessment of genetic interaction beyond gene pairs is a strong selling point and would be of interest of researchers in various research fields. Therefore, the manuscript is suitable for publication in Nat. Commun. However, I found that the text contains several ambiguous descriptions, and therefore have the following comments that need to be addressed publication.

1. Figure 3. In the text, 'Figure 3' is first cited in a section describing a prototype library on page 5 and kept cited while describing screening results with this prototype library, together with Supplementary Figures 6 and 7. However, Figure 3 and Supplementary Figure 7 show data with Inzolia library (Supplementary Figure 6 is unclear). These are very confusing. Figure presentation would be very similar between prototype and Inzolia libraries, but it would be clearer if the authors show the data in

separate figures.

2. Figure 3B. The authors wrote 'Quality control metrics met expectations (Figure 3B)'. How does this violin plot of log2 read counts explain QC metrics? What do the authors actually mean by is QC metrics?

3. Page 7. 'the one-component and two-component libraries yielded equivalent results (Sup. Fig. 7)'. I believe there is no such data in Sup. Fig. 7.

4. Page 7. 'THE whole-genome libraries'. Which libraries did the authors mention? Inzolia only or together with other libraries?

5. Figure 4H. There are 2 values that seem missing (ARAF_BRAF interaction). Or, are these omitted as they are zero?

6. Page 8. 'resulting in masking/positive genetic interactions'. As the legend to Supp. Fig. 8 says, single knockout of CCT complex and proteasome induced a severe fitness defect. In such case, single KO showed a fitness defect with the maximum level that the screen can detect (fitness=0 when fitness of wildtype = 1), and thus fitness loss by a second gene cannot be detected. Is 'masking positive interaction' applicable to these cases?

7. In Inzonia screens, single gene knockout will be done by expressing 4 crRNA from a single vector. The library contains 2 construct per single gene KO, but the same 4 crRNA are used in a different order in the In4mer array. Would this be a meaningful approach? Can they only serve as an in-sample duplicate (i.e. technical replicates)? It seems that the authors unnecessarily doubled the size of the single KO portion.

8. It would be appreciated if the authors discuss potential limitations of Inzonia screens. One potential difficulty that may be caused by multiple crRNA expression even for single gene KO is that a larger number of double strand breaks will be induced, compared to fewer crRNA/gRNA expression. Cancer genomes contain copy number/structural alteration, and as it has been shown before these sites show unusual behavior. Cells with the intact DNA repair, such as primary cells, may not be able to cope with the amount of DSBs induced by Inzonia library. There may be other limitations. It would be important to provide sufficient prewarning to those who are interested in using this library.

Reviewer #1 (Remarks to the Author):

In Anvar et al., the authors compare their combinatorial screening platform, which uses Cas12a, to other previous screens which used Cas9. They also describe a new library generated using the Cas12a system. The IN4MER platform is the state-of-the-art – that said I have a number of specific comments on this paper.

1. This paper doesn't have any supplementary tables or links to Github or FigShare. As I read the paper, I was looking for tables that had the gRNA sequences, the hits from the various screens performed, and the count files but I was unable to locate these. There are other elements of the analysis that I would really like to know – for example the identity of the 388 gene pairs that were scored as hits across all five multiplex perturbation platforms (Figure 1E). I would also like to know the hits called in each of the different screens. Because I don't have this information it makes it difficult for me to understand if the screen hits make biological sense. It would also be interesting to know which paralog families are SL. I can't find any of this information in the manuscript.

We find that it is important not just to share all data and analysis associated with our published work, but also to include links to those resources in the paper itself. Snark aside, we are embarrassed by this oversight as we have made everything available on figshare with a doi link at the end of the paper:

Data and Reagent Availability

The Inzolia library is available on Addgene as catalog #209551 and #209552.

All data and code for this study are available on Figshare at
[doi://10.6084/m9.figshare.24243832](https://doi.org/10.6084/m9.figshare.24243832)

2. In comparing screens, the authors don't appear to consider differences in experimental design. For example, each of the different screens have been sequenced to a different level of coverage, which might influence the ability to detect "hits". The screens were also performed at different timepoints. Collectively, these variables reflect aspects of screen set-up rather than the performance of the different Cas systems/vectors and the role they may play are not discussed.

We appreciate the reviewer's insightful comment. We have added the following paragraph to the results & discussion section:

We note that our approach does not consider the experimental designs used in each of the five paralog synthetic lethality studies. Differences in the set of chosen paralogs, library coverage, experimental timepoint, and sequencing depth, for example, might be sources of variance among each of the studies. We assumed that research teams applied best practices to their studies, but it is also possible that experimental design, rather than intrinsic platform robustness, dominates the within-study variation that we observed.

3. There are lots of messages in this paper. That the IN4MER platform is “better”, that Cas12a screens when performed in the way that the authors describe can be used to identify and explore paralogs as pairs and additional paralog gene sets etc. It feels to me that the authors are holding a biological story back or have plans to publish the screen results elsewhere. If that’s the plan, then the authors should just say that.

While we certainly believe the IN4MER platform is “better”, we intend for this to be the major story of the paper, and we believe in rapid publication of useful tools to the research community. We thank the reviewer for offering more credit than we deserve!

4. A library like the one described in the paper with multiple identical repeats might be expected to undergo recombination. Have the authors systematically examined this?

From the beginning of working with Cas12a we have been concerned about this possibility and thus have taken several steps to monitor and mitigate. First, we avoid the use of identical repeats, drawing on the characterization of alternative DR sequences that nevertheless maintain high activity with Cas12a (described in DeWeirdt - 2021, Nature Biotechnology, Figure 4). Second, were recombination to occur at a meaningful level, we would observe smaller products during PCR step of guide retrieval from genomic DNA. Here we show a gel of the PCR products from cells screened with the Inzolia library, showing clear bands of the proper size.

Notwithstanding the comments above the paper describes an advance that will be useful to the field.

Reviewer #2 (Remarks to the Author):

Anvar et al. generated a Cas12a crRNA platform, termed In4mer, where 4 crRNAs can be cloned into each vector molecule and tested their genetic interactions. Using this platform, they generated a new Cas12a library, Inzolia, targeting singles, pairs, trios and quads with approximately 50k clones,

substantially wider targets with a smaller size. Screens with this library identified expected genetic interactions in fitness and under drug treatment.

The 4-crRNA platform is an extension of the previous discovery by the authors (Nat Biotechnol 2021), but has made wider applications possible. In particular, assessment of genetic interaction beyond gene pairs is a strong selling point and would be of interest to researchers in various research fields. Therefore, the manuscript is suitable for publication in Nat. Commun. However, I found that the text contains several ambiguous descriptions, and therefore have the following comments that need to be addressed publication.

1. Figure 3. In the text, 'Figure 3' is first cited in a section describing a prototype library on page 5 and kept cited while describing screening results with this prototype library, together with Supplementary Figures 6 and 7. However, Figure 3 and Supplementary Figure 7 show data with Inzolia library (Supplementary Figure 6 is unclear). These are very confusing. Figure presentation would be very similar between prototype and Inzolia libraries, but it would be clearer if the authors show the data in separate figures.

We appreciate the reviewer's comments and we have clarified Figure 3C,D,E and Figure 4H to reflect which screen data were derived from the prototype and final Inzolia libraries. Supplementary Figure 7 accurately describes Inzolia data.

2. Figure 3B. The authors wrote 'Quality control metrics met expectations (Figure 3B)'. How does this violin plot of log2 read counts explain QC metrics? What do the authors actually mean by is QC metrics?

We thank the reviewer for this observation. We intended for all of Figure 3B-E to indicate screen quality controls. We have updated the text as follows:

We conducted screens in K562, and in A549, a *KRAS* lung cancer cell line with wildtype *TP53*, using standard CRISPR screening protocols (500x library coverage, 8-10 doublings). Array amplicons were sequenced using single-end 150-base Illumina sequencing. **Quality control metrics met expectations (Figure 3B-E). The library was well-sampled in each replicate (Figure 3B)**, and the abundance distributions of endpoint replicates were highly correlated (Figure 3C).

The remainder of this paragraph, and the following, discuss each panel in Figure 3 and are unchanged. We hope this clarifies our intent.

3. Page 7. 'the one-component and two-component libraries yielded equivalent results (Sup. Fig. 7)'. I believe there is no such data in Sup. Fig. 7.

The reviewer is correct. We have updated Supplementary Figure 7 to include the intended data for one- and two-component libraries and their comparison to the Humagne library screens in the same cell lines.

4. Page 7. 'THE whole-genome libraries'. Which libraries did the authors mention? Inzolia only or together with other libraries?

We thank the reviewer for identifying this point of confusion. We have updated the text as follows:

Our prototype and Inzolia whole-genome libraries target small paralog families as well as single genes.

5. Figure 4H. There are 2 values that seem missing (ARAF_BRAF interaction). Or, are these omitted as they are zero?

We thank the reviewer for catching this oversight. The ARAF_BRAF gene pair is indeed in the Inzolia library but not in the prototype. The updated Figure 4H clarifies which screens were done with which library, and the figure legend is updated as well:

*...H) Single, double, and triple knockout phenotype of RTK/MAP kinase pathway genes in all four cell lines. **White, target not in library.***

6. Page 8. 'resulting in masking/positive genetic interactions'. As the legend to Supp. Fig. 8 says, single knockout of CCT complex and proteasome induced a severe fitness defect. In such case, single KO showed a fitness defect with the maximum level that the screen can detect (fitness=0 when fitness of wildtype = 1), and thus fitness loss by a second gene cannot be detected. Is 'masking positive interaction' applicable to these cases?

We thank the reviewer for this insightful observation, and we have modified the paragraph as follows:

Likewise, interactions between essential genes are also a challenge to interpret. Both the core proteasome and the Chaperonin-Containing TCP1 (CCT) complex are composed of several weakly related proteins, which we target with three four-way constructs and numerous two-way constructs. Since both the proteasome and the CCT complex are universally essential to proliferating cells, knockdown of single subunits induces a severe fitness phenotype. Knockout of these genes in pairs or quads yields no additional phenotype, resulting in **what could be seen as** masking/positive genetic interactions in all four cell lines (Supplementary Figure 8). **However, when the expected double knockout fitness exceeds the dynamic range of the assay – e.g. when the sum of two single knockout log fold changes is more severe than any observed fold change in the screen – a more conservative approach is to consider these pairs to be untestable rather than positive interactions.**

7. In Inzolia screens, single gene knockout will be done by expressing 4 crRNA from a single vector. The library contains 2 construct per single gene KO, but the same 4 crRNA are used in a different order

in the In4mer array. Would this be a meaningful approach? Can they only serve as an in-sample duplicate (i.e. technical replicates)? It seems that the authors unnecessarily doubled the size of the single KO portion.

This is an excellent question. Our Supplementary Figure 6 shows the correlation between guide arrays containing the same gRNA sequences in different order and between technical replicates. It is clear that even the same gRNAs with different order still show considerable variation between technical replicates. Therefore, we believe it is worth having these technical replicates in the library.

8. It would be appreciated if the authors discuss potential limitations of Inzolia screens. One potential difficulty that may be caused by multiple crRNA expression even for single gene KO is that a larger number of double strand breaks will be induced, compared to fewer crRNA/gRNA expression. Cancer genomes contain copy number/structural alteration, and as it has been shown before these sites show unusual behavior. Cells with the intact DNA repair, such as primary cells, may not be able to cope with the amount of DSBs induced by Inzolia library. There may be other limitations. It would be important to provide sufficient prewarning to those who are interested in using this library.

We thank the author for highlighting this important point and we have added this text to the discussion:

While Cas12 exhibits advantages in multiplexing, its success relies on selecting a suitable biological model and ensuring optimal gRNA efficiency. Similar to Cas9, a potential challenge in screening with Cas12 is the induced double-strand breaks, triggering a DNA damage response and subsequent cell cycle arrest. Previous studies have highlighted that the number of loci targeted by CRISPR, particularly those spanning chromosomes, can result in gene-independent fitness loss, potentially leading to a higher rate of false-positive identification of undesired cell-essential genes (Aguirre,2016,Cancer Discovery). Although Berg et.al (Berg, 2018, Nucleic Acids Research) has revealed the limited differences between one cut and four cuts resulting to H2AX changes and most cancer cell lines exhibit tolerance to some extent of DNA damage response, the potential for high copy numbers and off-target effects induced additional breaks may compromise accuracy.

Moreover, the Inzolia library is designed to address this concern. It incorporates non-targeting and intergenic controls, allowing us to assess gene-independent changes in fitness. Notably, positive enrichments have been observed between non-targeting control/EGFR and intergenic/nonessential controls in different cells (Figure), but it is similar to CRISPR-Cas9 screening (Goncalves, 2021, Genome biology). Consequently, we have adopted non-essential controls as a baseline in our analysis to mitigate potential false-positive hits. Our quality control analyses indicate that the false discovery rate (FDR) for Cas12 is comparable to Cas9, even when utilizing four guides in an array, thereby not significantly increasing the risk of DNA damage-induced depletion. While our studies have not investigated DDR sensitive cell lines like primary cells, it is worth investigating their response on our platform for future research. Notably, a significant deviation in the distribution of non-essential elements from the norm may imply independent gene-related anti-proliferative effects. Moreover, different biological models might require optimized delivery systems for in4mer platform, especially in primary cells and in vivo.

Reviewers' Comments:

Reviewer #1:

Remarks to the Author:

This is an important paper that moves the field forward by describing new methods for screening multiple genes, and also a new approach for CRISPR screen analysis. That said I do find it hard to follow the work done in the manuscript. For example, the supplementary tables are still not cited in the text. I can find supplementary Table 4.1, 4.2 and 5 on Figshare but there are no legends for these tables and I can't find Supplementary Tables 1-4 (if these exist). The 388 gene pairs called across the screens are in a file called "hits_388.csv" but there are no column descriptors and the authors have not pointed readers at this table in their text. I have several other specific points.

1. Why Inzolia? I am aware this is a grape variety but is the library called Inzolia because it's light and zesty? Is it "IN4MER" because up to 4 guides can be included or because IN4MER is close to informer? I am not suggesting that boring names should be used just that the authors define the "jargon" they have coined so that others, particularly those who have not followed the Broad's naming trend, can understand what is being described and why.
2. "We identified a total of 26 gene pairs that meet these criteria" What are these pairs? How does a reader look at them?
3. "We assumed that research teams applied best practices to their studies, but it is also possible that experimental design, rather than intrinsic platform robustness, dominates the within-study variation that we observed." I think this is fair enough but the "best practice" for screens of this type are still being established (as evidenced in this paper) and even the authors describe methodological improvements in this work when compared to their prior study Dede et al.,. Similarly, Cas9 gRNA design has improved significantly over the last 2 years with tools like VBC being released.
4. A549, a KRAS mutant lung cancer cell line with wildtype TP53
5. It's probably worth the authors looking in or similar so they use the correct cell line names. For example A-549. Mel JuSo not Meljuso. Snark – gene names in italics.
6. In places citations are lacking. For example, DepMap (the authors own work) is mentioned but never cited.

Overall, I think the work is an advance but it could really do with a bit more attention to polish the manuscript and to make the data and analysis more useful for others.

Reviewer #2:

Remarks to the Author:

The authors have addressed all my concerns. The manuscript is now suitable for publication.

Response to reviewers, 6 Feb 2024

REVIEWER COMMENTS

Reviewer #1 (Remarks to the Author):

This is an important paper that moves the field forward by describing new methods for screening multiple genes, and also a new approach for CRISPR screen analysis. That said I do find it hard to follow the work done in the manuscript. For example, the supplementary tables are still not cited in the text. I can find supplementary Table 4.1, 4.2 and 5 on Figshare but there are no legends for these tables and I can't find Supplementary Tables 1-4 (if these exist). The 388 gene pairs called across the screens are in a file called "hits_388.csv" but there are no column descriptors and the authors have not pointed readers at this table in their text. I have several other specific points.

We thank the reviewer for these important comments. The entire figshare repository has been reorganized, with well-labeled code and data files associated with each figure. Importantly, the Supplementary Tables 1-9 are appropriately titled and are referenced in the main text. To wit:

Supplementary Table Legends

Supplementary Table 1: dLFC and Cohen's D for five published paralog studies

Supplementary Table 2: 388 Identified synthetic lethals, 26 synthetic lethals and gold standard set

Supplementary Table 3: Raw read counts for 7mer screens

Supplementary Table 4: Prototype library sequence

Supplementary Table 5: Inzolia library sequence

Supplementary Table 6: Raw read counts for both Prototype and Inzolia library screens

Supplementary Table 7: Log fold change by gene for both Prototype and Inzolia library screens

Supplementary Table 8: DrugZ score for MELJUSO screen with MEK inhibitor selumetinib

Supplementary Table 9: GSEA result for MELJUSO screen with MEK inhibitor selumetinib

1. Why Inzolia? I am aware this is a grape variety but is the library called Inzolia because it's light and zesty? Is it "IN4MER" because up to 4 guides can be included or because IN4MER is close to informer? I am not suggesting that boring names should be used just that the

authors define the “jargon” they have coined so that others, particularly those who have not followed the Broad’s naming trend, can understand what is being described and why.

2. “We identified a total of 26 gene pairs that meet these criteria” What are these pairs? How does a reader look at them?

All figures are now associated with an underlying data file, per Nature Communications editorial requirements, and all results from the paralog meta-analysis are included in Supplementary Table 2.

3. “We assumed that research teams applied best practices to their studies, but it is also possible that experimental design, rather than intrinsic platform robustness, dominates the within-study variation that we observed.” I think this is fair enough but the “best practice” for screens of this type are still being established (as evidenced in this paper) and even the authors describe methodological improvements in this work when compared to their prior study Dede et al.,. Similarly, Cas9 gRNA design has improved significantly over the last 2 years with tools like VBC being released.

We agree.

4. A549, a KRAS mutant lung cancer cell line with wildtype TP53

We assume the reviewer would like this text added; we have added it.

5. It’s probably worth the authors looking in or similar so they use the correct cell line names. For example A-549. Mel JuSo not Meljuso. Snark – gene names in italics.

We have edited cell line names to be consistent with prior Springer/Nature publications in which these authors have described work with these cells (e.g. DeWeirdt et al, Nature Communications, 2020).

6. In places citations are lacking. For example, DepMap (the authors own work) is mentioned but never cited.

We have added the reference to Tsherniak 2017.

Overall, I think the work is an advance but it could really do with a bit more attention to polish the manuscript and to make the data and analysis more useful for others.

Reviewer #1 (Remarks on code availability):

I am an R programmer and the code is in Python. That said the code is there but I can't see all the files i need to execute it. You could ask the authors to provide it as fully executable. The code could also be improved with a fuller description of each step. For example at present the documentation says things like "# clean the data" but there is no details of what exactly the code chunk does and why. It would only take a few lines to make it accessible to others.

Reviewer #2 (Remarks to the Author):

The authors have addressed all my concerns. The manuscript is now suitable for publication.

Reviewers' Comments:

Reviewer #1:

Remarks to the Author:

The paper is improved. I will leave it to the editorial/production process to make sure that all of the various tables/files/code is documented and referenced.

The work is important and I look forward to seeing the method deployed by others.